# Oral Pre-Exposure Prophylaxis Innovative Interventions among Adolescent Girls and Young Women in South Africa: A Protocol Paper

**DOI:** 10.3390/mps7050077

**Published:** 2024-09-29

**Authors:** Lerato Lucia Olifant, Edith Phalane, Refilwe Nancy Phaswana-Mafuya

**Affiliations:** Faculty of Health Sciences, South African Medical Research Council/University of Johannesburg (SAMRC/UJ) Pan African Centre for Epidemics Research (PACER) Extramural Unit, Johannesburg 2006, South Africa; leratoo@uj.ac.za (L.L.O.); edithp@uj.ac.za (E.P.)

**Keywords:** pre-exposure prophylaxis, adolescent girls and young women, HIV/AIDS, COVID-19, innovative interventions, South Africa

## Abstract

Although South Africa was the first country to register and roll out oral pre-exposure prophylaxis (PrEP) biomedical human immunodeficiency virus (HIV) prevention intervention in sub-Saharan Africa (SSA), its uptake remains low, particularly among adolescent girls and young women (AGYW). The uptake of PrEP may have worsened during the Coronavirus disease 2019 (COVID-19) pandemic. Some innovative interventions to improve PrEP uptake among AGYW have been implemented. This study aims to evaluate the effectiveness of PrEP innovative interventions implemented during COVID-19 towards reducing the risk of HIV infection among AGYW in South Africa. An exploratory, descriptive design will be conducted to carry out four study objectives. Firstly, to carry out a systematic review of innovative PrEP interventions implemented during COVID-19 in SSA countries. Secondly, to conduct a stakeholder analysis to identify PrEP stakeholders and interview them on their views on the implemented interventions. Thirdly, to assess the implementation outcomes of the innovative interventions using document reviews and Consolidated Framework for Implementation Research. Fourthly, to develop a framework for an improved PrEP service delivery among AGYW. Qualitative data will be captured in ATLAS.ti software (Technical University, Berlin, Germany) version 23 and analysed via thematic analysis. A statistical software package (STATA) version 18 (College Station, TX, USA) will be used to capture quantitative data and analyse them via descriptive analysis. The generated evidence will be used towards the development of framework, guidelines, and policies to strengthen the uptake of, scale-up, and adherence to PrEP among AGYW.

## 1. Introduction

Significant progress has been made in controlling the burden of the human immunodeficiency virus (HIV); however, its incidence rates remain high, particularly among key populations, women, and other priority populations [1]. Globally, an estimated 4900 new HIV infections are reported weekly among adolescent girls and young women (AGYW) between the ages of 15 and 24 years [2]. In 2022, AGYW accounted for more than 77% of new infections in sub-Saharan Africa (SSA) [2], whereas in South Africa they constituted about 37% of all new HIV infections [3]. Interventions aimed at reducing the high incidence rates have been widely implemented. South Africa can be acknowledged for the substantial investments in these HIV interventions, which include the scaling up of education on and access to HIV testing, antiretroviral therapy (ART), and oral pre-exposure prophylaxis (PrEP), as well as the implementation of the universal test and treatment [4]. Oral PrEP is a biomedical HIV prevention and one of the most effective tools that have been implemented specifically, among AGYW [5].

Daily oral PrEP is a combination of two or more drugs like tenofovir and emtricitabine [6] and is only effective when taken as prescribed [7]. Although South Africa was the first country to register and roll out oral PrEP in SSA [8], its uptake remains low, particularly among AGYW [9,10,11]. Several barriers to PrEP uptake and adherence have also been indicated in other SSA countries like Kenya. These barriers include lack of access, low social support for users from peers and community, and inadequate information about PrEP [12,13], highlighting the need to evaluate PrEP interventions implemented for these population groups [14]. Moreover, other factors such as individuals’ awareness of the benefits of PrEP in reducing HIV infections also contribute to the overall care continuum. For instance, one study in Kampala, Uganda, on the uptake and adherence to PrEP among AGYW, found individual factors like knowledge of PrEP, HIV risk perceptions, and doubting the efficacy of PrEP play a significant role in the non-adherence rates [15]. Furthermore, the issue with early discontinuation is associated with the side effects of the prescribed PrEP medication. For example, some of the reasons for discontinuation among AGYW were the experienced side effects that did not subside, as well as the matter of swallowing PrEP pills due to their size [16].

The emergence of the Coronavirus disease 2019 (COVID-19) pandemic with its treatment protocols, prevention measures, and lockdown requirements may have exacerbated the uptake and adherence of PrEP among AGYW [17]. For instance, one South African study on the impact of COVID-19 on HIV prevention and treatment services among key populations showed a significant decrease in service engagement, which included declines in the number of people accessing HIV testing, treatment, and PrEP during the onset of the lockdown period [18]. Additionally, another study evaluated PrEP retention and prescriptions during the COVID-19 lockdown period in South Africa among pregnant women aged ≥ 16 years. The above-mentioned study found that missed PrEP visits were increased to 63% at the 1-month visit and 55% at the 3-month visit, during the lockdown period [19].

In controlling these service declines, suitable guidelines for developing and implementing innovative interventions for service continuity were provided by the World Health Organization (WHO) [20]. Innovative interventions are defined as novel or creative ways that can generate more impacts for more people [21]. Several innovative PrEP interventions like using virtual options for client initiations, refills and check-ins, decentralizing PrEP delivery, and moving to multi-month dispensing have been introduced [17,22]. These innovative PrEP interventions need to be evaluated as the 2023–2028 National Strategic Plan (NSP) on HIV, tuberculosis (TB), and sexually transmitted infections (STIs) emphasizes the urgent need for innovation [1]. As with other countries, South Africa continued with essential HIV services during the lockdown period to maintain its target; however, the effects of the COVID-19 pandemic on these services at the primary healthcare level remain unclear in South Africa [23]. Additionally, there are limited data regarding the successes and challenges of these interventions [24]. As a result, the current study will primarily focus on South Africa to evaluate these interventions in the long term for future pandemic preparedness. Furthermore, the findings will report on the feasibility, acceptability, and sustainability of these interventions, to inform the development of a framework for the improvement in service delivery.

### Research Questions

The proposed study will answer the following questions:○What is the status quo regarding innovative interventions implemented during COVID-19 towards improving the uptake of PrEP outcomes among AGYW in sub-Saharan Africa?○What are the roles of stakeholders involved in the implementation of innovative interventions towards improving PrEP uptake and outcomes among AGYW in South Africa?○What are the stakeholder views regarding types, places, and how PrEP innovative interventions have been implemented among AGYW in South Africa?○Were the implementation outcomes of PrEP innovative interventions among AGYW realized?

## 2. Methods

The proposed study will conduct an exploratory, descriptive design to address its four objectives, which include a systematic review method (objective 1); PrEP stakeholder analysis and qualitative in-depth interviews (objective 2); document reviews and semi-structured interviews to assess the implementation outcomes of PrEP innovative interventions (objective 3); and triangulating all the research methods applied to develop an improved PrEP service delivery approach (objective 4). The methods for each objective are discussed in the subsequent sections.

### 2.1. Method for Objective 1: To Conduct a Systematic Review to Identify and Describe the Innovative Interventions Implemented (e.g., Types, Places, Period, and How) during COVID-19 towards Improving the Uptake of PrEP among AGYW in SSA

The proposed review will be conducted in line with the preferred reporting items for systematic reviews and meta-analyses (PRISMA) guidelines [25] and is registered in the International Prospective Register of Systematic Reviews (PROSPERO) [CRD42023439020]. A comprehensive search will be conducted for both the published and grey literature. From this literature search, summaries of findings on innovative interventions implemented during COVID-19 towards improving PrEP outcomes among AGYW will be written. Furthermore, all the processes including the results, selection of full-text articles, as well as the included studies in the final systematic review will be presented using the PRISMA flow diagram as displayed in Figure 1.

Inclusion and exclusion criteria:

The inclusion and exclusion criteria will be guided by the Population Intervention Comparison Outcomes (PICO) framework. Table 1 below illustrates the inclusion criteria of the review:

Exclusion Criteria

All studies that did not meet the inclusion criteria will be excluded, for instance, those of meta-analyses, systematic reviews, and other types of secondary research. Other exclusion criteria for this review will include (1) studies not about PrEP; (2) not conducted in SSA; (3) not involving AGYW; (4) not describing PrEP interventions; (5) not describing or measuring PrEP outcomes or outcomes of the interventions; (6) no specific interventions and no outcomes; and (7) letters, editorials, and commentaries.

Search strategy and database search

The PICO framework, illustrated above in the inclusion criteria section and the Medical Subject Heading (MeSH) terms, will guide a comprehensive search strategy for this review. For literature searches, PubMed, Google Scholar, Scopus, and MEDLINE will be used.

Additionally, a manual search will also be conducted from websites like the Pre-Exposure Prophylaxis–https://www.aidsmap.com/about-hiv/pre-exposure-prophylaxis-prep (1 March 2024); and https://www.cdc.gov/hiv/risk/prep/index.html (1 March 2024)to ensure that more information from significant health organizations reports is retrieved. Moreover, the inclusion of external websites outside SSA like the Centers for Disease Control and Prevention (CDC) will be important in enabling the authors to secure a comprehensive overview of PrEP interventions since they were first approved in the United States of America. Also, the website can allow one to search for specific topics of interest. Lastly, references to the retrieved articles will be searched to ensure that any potential studies meeting the inclusion criteria can be included. Furthermore, a preliminary search via the PubMed database was conducted using the combinations of medical terms based on the aim of the review; these are illustrated in Table 2 below.

Quality appraisal

The quality appraisal will be ensured by using the Critical Appraisal Skills Programme Systematic Review (CASP SR) checklist (Appendix A), which investigates the validity, precision, and generalizability of the research [26]. Two team members will independently evaluate the complete text and abstract of the qualifying articles using the inclusion and exclusion criteria, as well as a predetermined and agreed-upon score criterion for each evaluation. In case of differences, a third member will give input and make the final decision based on the reviews of the two members.

Data extraction and evidence of synthesis

Two independent reviewers will extract eligible articles using a developed data extraction tool (Appendix B). The tool will be developed according to the review’s aim and on the in-depth discussions between the reviewers to ensure that all the significant errors are minimized. The included headings of the tool will be the study and author/s names, year of publication, country, study design, study population, intervention type, a brief description of the intervention, and key findings. Furthermore, on the interventions, additional information will be extracted, for instance, the frequency, duration of the intervention, uptake, accessibility, feasibility, and sustainability of these innovative PrEP interventions. Lastly, author papers will be contacted in case of missing information/data. Moreover, references will be managed by RefWorks software version 6.0. A narrative summary of the results will be developed, and tables will be used to show specific details of the analysis.

### 2.2. Methods for Objective 2: To Determine Stakeholders’ Views Regarding PrEP Innovative Interventions Implemented among AGYW during COVID-19 in South Africa, and Successes and Improvements That Can Be Made

This objective involves conducting a stakeholder analysis and in-depth interviews with stakeholders.

#### 2.2.1. Stakeholder Analysis

A stakeholder analysis will be performed to identify stakeholders and gather information about their interests, roles, responsibilities, influence, networks, and resources regarding PrEP uptake among AGYW. Understanding actors’ interests and responsibilities makes it feasible to know why and how particular interventions were implemented. As described below, a six-step process adapted from the UNAIDS/Rwanda [27] will be followed.

Step 1: Defining the study’s objective

The study’s objective is to explore stakeholders’ views on the implemented innovative PrEP interventions during the lockdown period. The researcher will firstly commence with consultations with various operational managers from the primary healthcare facilities where these interventions were implemented using gatekeepers’ approvals. During these consultations, the researcher will explain the aim of the study and the requirements thereof. From these discussions, the operational managers will have an overview of the study and be in a position to refer the researcher to the relevant stakeholders/participants who were involved in innovative PrEP interventions during the COVID-19 pandemic. Also, those who have been working in the field of PrEP services even before the lockdown period for possible participation in the study.

Step 2: Stakeholder identification

The second step will be the identification of the stakeholders who are involved in the HIV PrEP services. The researcher will ask the operational manager to schedule a meeting where the research information will be discussed with the relevant stakeholders. During this meeting, the researcher will compile a list that will be circulated among the relevant stakeholders, which will include information on their various positions in PrEP services. From this list, the researcher will be able to identify and categorize these stakeholders according to their roles and responsibilities in implementing innovative PrEP interventions among AGYW.

Step 3: Stock-taking of the current relationship

This third step is significant in assessing the stakeholder relationships in the implementation of PrEP innovative interventions among AGYW according to their various roles. To achieve this goal, a mapping tool will used to score these relationships. The tool is a 10-point scale and categories are labelled as follows: 1–2 = No Interaction, 3–4 = Rare, 5–6 = Intermittent, 7–8 = Regular, and 9–10 = Constant and Consistent. Then, each stakeholder will be expected to use the above-mentioned tool to score each other’s relationships.

Step 4: Determining the resource-based influence

This fourth step focuses on direct resources used in the implementation of PrEP innovative interventions, like extra time, money, and HIV prevention packages for AGYW that were invested by the stakeholders. This step will be determined through discussions where stakeholders will point out any resources employed during the COVID-19 lockdown period to ease service continuity. The process will be beneficial to assist in determining the challenges faced during the implementation of the interventions.

Step 5: Determining the non-resource-based influence

This fifth step is about the assessment of the non-resource-based influence in the implemented interventions. For example, the roles that each stakeholder played in these interventions will be assessed by their various responsibilities/roles in reaching out to AGYW during the pandemic to improve access to services. Stakeholders will be asked to score one another using a 10-point scale where 1 will be regarded as a non-resource-based influence and 10 as a high influence.

Step 6: Reviewing and revising the collaboration map

This last step is concerned with the reviewing of the collaboration relationship between the stakeholders in the development and implementation of innovative PrEP interventions. This step will also assist stakeholders in viewing their roles towards improving PrEP uptake among AGYW and also view how they can strengthen their relationship to achieve the implementation outcomes.

#### 2.2.2. In-Depth Interviews with Stakeholders

*Study design:* An exploratory study design will be used.

This design is used when the researcher has limited topic knowledge [28]. It is suitable for this study because little is known about the innovative PrEP interventions implemented among AGYW during COVID-19.


*Sample:*


A purposive sampling method will be employed to select 15–20 stakeholders who were involved in the development and implementation of the PrEP innovative interventions among AGYW. Since most services were rendered from clinic facilities to community settings during the pandemic, recruitment will be undertaken from key informants who were involved in these community outreach activities. These key informants must be 18 years and above and have been involved in developing and implementing innovative PrEP interventions among AGYW. Table 3 displays detailed inclusion and exclusion criteria of the study participants.


*Participants recruitment:*


As PrEP services are often provided by public (government) healthcare facilities, recruitment procedures will be conducted in these local clinics. The recruitment will be performed both directly and through the operational manager’s referral network, where the researcher will engage with the clinic operational manager to explain the study and be physically linked to the relevant PrEP stakeholders for participation. The operational manager must be an individual who works closely with the stakeholders within the healthcare facility, is experienced in PrEP services, and can assist with recruitment processes. After the researcher has met with the operational manager, prospective participants will be physically invited to an information session in the facility where the study details will be explained thoroughly. During the information session, both the study information letter and the informed consent forms will be distributed to participants to ensure the voluntary nature of the study. The researcher will also compile a contact list of all potential participants to be contacted for the interview schedule either through WhatsApp messages or phone calls depending on their preferences.


*Data collection:*


Before the actual data collection, potential participants will be telephonically contacted for the interview schedules. One-on-one in-depth interviews will be used as a data collection tool. Furthermore, the researcher will request a private room in the primary healthcare facility where interviews will be conducted.


*Data analysis:*


The interview transcript will be analysed using thematic content analysis. After the transcription of the audio, the information will be imported into ATLAS.ti software (Technical University, Berlin, Germany) version 23. All the field notes and interview transcripts will be analysed with thematic content analysis through which themes within a dataset are identified, analysed, and reported. These results will provide insightful information on the stakeholder characteristics towards improving PrEP among AGYW in South Africa.

### 2.3. Methods for Objective 3: To Assess the Implementation Outcomes of PrEP Innovative Interventions among AGYW in South Africa


*Study design:*


Exploratory–descriptive design will be employed to conduct both document review and stakeholder interviews regarding existing PrEP innovative interventions.


*Inclusion and exclusion criteria:*


The document review will include published (policies in the implementation of innovative interventions) and unpublished documents (meeting minutes, implementation plans, monitoring, and evaluation reports) on innovative PrEP interventions. The documents should be written in English. Additionally, unpublished documents such as routine PrEP programme data from the facility will be included. Interviews will be undertaken with HIV PrEP service providers, and intervention implementers, who must have been involved in PrEP services for at least one year in the field irrespective of their gender, and they must be 18 years and above.


*Area of study:*


The study will be conducted in the Dr Kenneth Kaunda District (Dr KK), North-West Province, South Africa. The Dr KK District is one of the four municipalities, namely, Bojanala Platinum, Ngaka Modiri Molema, and Dr Ruth Segomotsi Mompati, in the North-West Province [29]. The Dr KK District was selected because it is one of the Districts that has responded to COVID-19 by implementing several interventions related to the provision of water tanks as well as the Decentralizing Service Delivery feeding programmes [30]. Moreover, the first author (LLO) had previously engaged with the health officials in the North-West Department of Health (NWDoH) regarding Districts that were involved in the implementation of innovative PrEP interventions during the COVID-19 lockdown period. The District was mentioned as one municipality that had been involved in HIV service continuity irrespective of the COVID-19 disruptions.


*Data collection:*


Published documents will be searched from reports, websites, and any pertinent sources. A data extraction tool guided by the CFIR constructs will be used to identify relevant information (Appendix C) on the reviewed documents. These will include constructs like, acceptability, adoption, appropriateness, cost, feasibility, fidelity, penetration, and sustainability [31]. There will also be semi-structured interviews guided by the above-mentioned constructs, which will involve 15–20 stakeholders to assess the implementation outcomes of the innovative PrEP interventions. The stakeholders will be invited for the one-on-one semi-structured interviews scheduled at their convenience. Before the commencement of the interviews, the participants will be asked to voluntarily sign an informed consent form. The interviews will be conducted in a safe and private space chosen by the participants within the facilities. The interviews will last for 30–45 min and will be audio-recorded. Furthermore, interviews will be conducted in any/mixed languages (English or Setswana) that the participants are most comfortable with.


*Data Analysis:*


The ATLAS.ti software version 23 will be used to capture the interview transcripts. Captured data will include both open-ended and closed-ended questions. The qualitative data will be analysed using thematic content analysis. A descriptive analysis will be used to analyse any closed-ended information using a statistical software package (STATA) version 18 (College Station, TX, USA). Descriptive analysis will be conducted to provide a summary of the frequencies.

### 2.4. Methods for Objective 4: To Develop a Framework for an Improved PrEP Service Delivery Approach among AGYW


*Triangulation:*


Various findings from the first 3 objectives will be systematically analysed and triangulated to develop a framework for an improved PrEP service delivery approach for AGYW. Figure 2 below unpacks the processes of triangulations. Additionally, these processes will first involve the analyses of each finding separately to maintain the integrity of all findings. Secondly, all findings will be compared and contrasted as well as taking into consideration existing literature to elicit key findings/themes that will be part of the components of the framework. Lastly, these findings will be synthesized, interpreted, and discussed in line with the study’s objectives.

## 3. Ethical Procedures

Ethical guidelines will be adhered to throughout the research process. The study has obtained ethical approval (REC-2435-2023) from the University of Johannesburg Research Ethics Committee (UJREC), and the North-West Department of Health (NWDoH) for data collection. Since human participants will be used in the study, several principles will be adhered to, enhancing the norms and standards of ethics as well as aligning with the guidelines of the Protection of Personal Information Act (POPIA). Before taking part in the study participants will be required to give written study informed consent and POPIA consent for the collection of personal information. Furthermore, study participants will be ensured that participation should be strictly voluntary, and they are allowed to withdraw from the study.

Collected data will only be used for study purposes and participants will be respected throughout the research project. For the document reviews, precautionary measures will be adhered to, by removing any identities, the dataset will not contain information like names, identity, passport numbers, contact details, as well as physical addresses aligning with the POPI Act. The researcher will ensure that the benefits outweigh the unforeseen risks of the study. The semi-structured interviews will be conducted in a safe environment like their workplace where participants will be comfortable in participating. All the interview transcripts and field notes will only be accessed by the parties involved in the study. The researcher will ensure that pseudonyms are used for each participant and no identifiable information will be found in the interview transcripts. A secure filing system will be in place with a locking mechanism where all the hard copies (informed consent forms, field notes, and interview transcripts) will be saved.

## 4. Expected Results

The current study forms part of the doctoral study by the first author (LLO), which aims to evaluate the effectiveness of PrEP innovative interventions implemented during COVID-19 towards reducing the risk of HIV infection among AGYW in South Africa to inform the development of a framework for the improvement of service delivery. The aim is to employ an exploratory descriptive design to evaluate the success and challenges of these interventions. This evaluation will provide a comprehensive understanding of other innovative approaches employed for service continuity. As with other key populations, AGYW are required to be retained within the PrEP care continuum to achieve HIV infection control by 2030. The expected results from the current study will contribute to enhancing the PrEP service delivery among AGYW and assist in achieving the above-mentioned goal. Additionally, the study findings will highlight the most utilized popular interventions, which can be further evaluated and improved for better PrEP services among AGYW. Furthermore, the study will unpack the feasibility, accessibility, and sustainability of the implemented interventions to prepare for future pandemics.

### Strength and Limitations

The current study will contain some limitations such as the selection of participants and recall bias about the COVID-19 lockdown period events. However, the significant strength of employing a mixed methods approach will address this. The combination of both qualitative and quantitative research will provide a comprehensive understanding of the successes and challenges of the innovative PrEP interventions implemented during the lockdown period. Furthermore, the flexibility of the research approach will enable the researchers to explore valuable insight into the COVID-19 events.

## 5. Conclusions

In conclusion, with this assessment, we intend to guide healthcare professionals and policymakers in scaling up and evaluating innovative PrEP interventions among AGYW. The findings will also provide a comprehensive understanding of the facilitators and barriers to the implementation of these innovative interventions. Furthermore, the feasibility, accessibility, and sustainability of these interventions will also be outlined in the findings.

## Figures and Tables

**Figure 1 mps-07-00077-f001:**
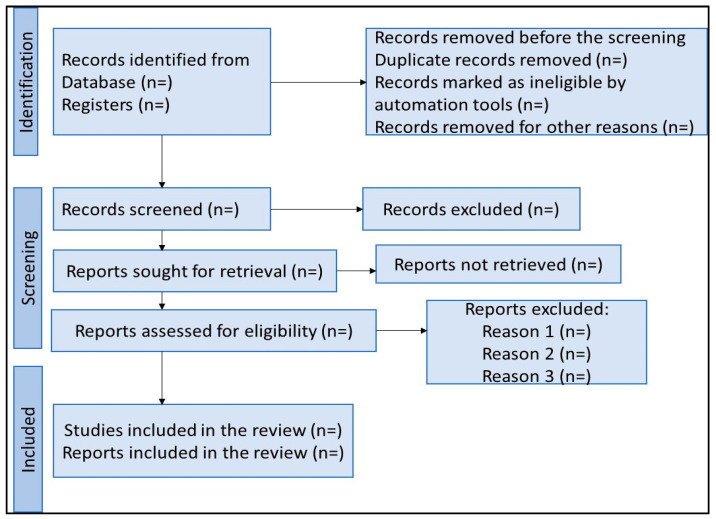
PRISMA flow diagram adapted from Page et al. [25].

**Figure 2 mps-07-00077-f002:**
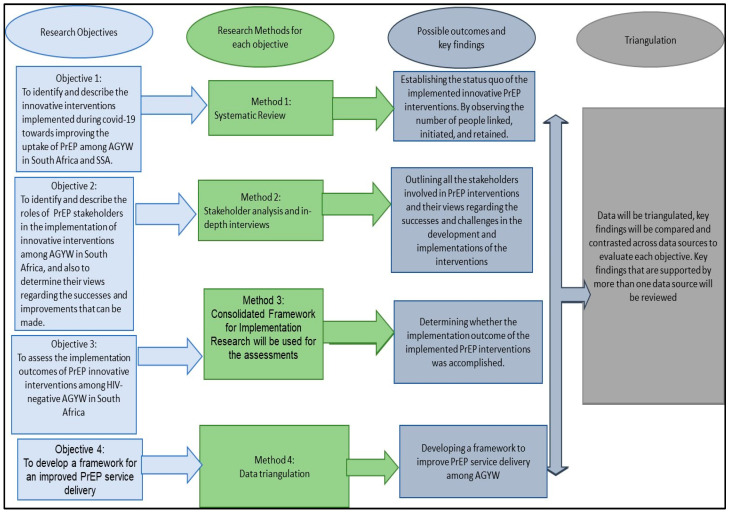
Summary of data sources that will be used in the study.

**Table 1 mps-07-00077-t001:** The inclusion criteria using the Population Intervention Comparison.

Aspect	Description
Population	Adolescent girls and young women between the ages of 15 and 24 years Residing within sub-Saharan African countries Involved in innovative PrEP interventions during the COVID-19 lockdown period
Intervention	Innovative PrEP interventions implemented during the COVID-19 lockdown period from 5 March 2020 to 30 June 2020 The interventions will be described in terms of the target population, types, and content of interventions, duration of the intervention, outcomes of the interventions, PrEP outcomes, key findings, and effectiveness of the interventions
Comparison	The review will compare interventions implemented before and during COVID-19 among AGYW.
Outcomes	The review will assess how feasible, accessible, and sustainable these interventions were in various countries in SSA. Some of the PrEP outcomes that will be assessed include the uptake and adherence rates among the target population. As well as the success and challenges of the implemented PrEP interventions

**Table 2 mps-07-00077-t002:** Sample search strategy in electronic databases.

Search	Query
# 1	Search: (AGYW OR adolescent female OR youths OR teens OR female adolescents OR teenagers OR young adults) “AGYW”[All Fields] OR (“adolescent”[MeSH Terms] OR “adolescent”[All Fields] OR (“adolescent”[All Fields] AND “female”[All Fields]) OR “adolescent female”[All Fields])
# 2	Search: (Pre-Exposure Prophylaxis OR PrEP OR methods OR delivery health care OR community distribution) “Pre exposure prophylaxis”[MeSH Terms] OR (“pre-exposure”[All Fields] AND “prophylaxis”[All Fields]) OR “pre-exposure prophylaxis”[All Fields] OR (“pre”[All Fields])
# 3	Search: (coronavirus disease 2019 OR SARS-CoV-2 infections OR COVID-19 pandemic OR 2019 Novel coronavirus disease OR 2019-ncov disease) “COVID-19”[Supplementary Concept] OR “COVID-19”[All Fields] OR “coronavirus disease 2019”[All Fields] OR “COVID-19”[MeSH Terms] OR “COVID-19”[Supplementary Concept] OR “COVID-19”[All Fields] OR “SARS-CoV-2 infections”[All Fields] OR “COVID-19”[MeSH Terms]
# 4	Search: (Sub-Saharan Africa OR SSA OR Africa OR South Africa OR Nigeria OR Congo OR Ethiopia OR Lesotho OR Botswana OR Cameroon OR Zimbabwe OR Rwanda OR Ghana OR Uganda OR Kenya) “Africa south of the Sahara”[MeSH Terms] OR (“Africa”[All Fields] AND “south”[All Fields] AND “Sahara”[All Fields]) OR “Africa south of the Sahara”[All Fields] OR (“sub”[All Fields] AND “Saharan”[All Fields] AND “Africa”[All Fields])
# 5	# 1 AND # 2 AND # 3 AND # 4

**Table 3 mps-07-00077-t003:** Inclusion and exclusion criteria.

Participants	Inclusion Criteria	Exclusion Criteria
► HIV PrEP service providers ► Intervention Implementers ► HIV PrEP intervention coordinators	► 18 years and above ► Stakeholders who were involved in HIV PrEP services during COVID-19 ► Stakeholders who were involved in HIV PrEP services before the pandemic ► Willingness and availability to participate in the study and to consent to audio record interviews ► Living within the Dr KK District	► Stakeholders who were not involved in HIV PrEP service during COVID-19 ► Providers and implementers who will not be able to commit to take part in the study ► Residing outside the Dr KK District

HIV—human immunodeficiency virus, PrEP—pre-exposure prophylaxis, COVID-19—Coronavirus disease 2019, Dr KK—Kenneth Kaunda.

## Data Availability

Not applicable.

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
