# Peer review of "Oral Pre-Exposure Prophylaxis Innovative Interventions among Adolescent Girls and Young Women in South Africa: A Protocol Paper"

_mps, 2024, doi:10.3390/mps7050077_

Round 1

Reviewer 1 Report

Comments and Suggestions for Authors

The authors described a beautiful protocol paper to study oral PrEP prophylaxis among AGYW in Sub-Saharan Africa. I have no comments to make on this manuscript. It is very well written. I accept this manuscript in this present form.

Author Response

Dear Reviewer,

Thank you for taking the time to review the manuscript, 

We appreciate the provided comments.

Best regards,

Reviewer 2 Report

Comments and Suggestions for Authors

This is a well written protocol. No additional comments

Author Response

(The authors gave the same response as above.)

Reviewer 3 Report

Comments and Suggestions for Authors

This study protocol paper needs more work. The overall premise of the study is interesting, however the paper as it is currently written needs much work to be ready for publication. 

Intro:

The authors appear to be interested in the PrEP care continuum among Black AGYW.  It would be ideal to 1) explicitly state that and 2) provide data on what the current PrPE care continuum looks like in this population.

The authors have chosen to use the COVID-19 pandemic (and the possibility that it will re-occur) as the impetus for doing this work. This makes this article have less importance. What they should focus on instead is how the COVID-19 pandemic may have changed AGYW perceptions and willingness to use PrEP (based on how COVID vaccines and misinformation during the pandemic were handled).

The systematic review that will be done appears to compare South African to SSA. More comparison is needed in the background section to make this method make sense.

METHODS:

What type of mixed methods study is this? Which strand, if any, is given greater weight? The order of quant vs quant should be mentioned in your type of study? Is this exploratory or confirmatory?

Spell out all acronyms with first use.

Figure 1 is not necessary since you have already mentioned you are following PRISMA guidelines. 

Inclusion/exclusion criteria- I thought this was focused on AGYW. Your are including AGYW 18 years old and older. Why is there not an upper range of age so that you are not also including women. If you want to include all women, please restructure your entire paper to make it relevant to women of all ages.  Define "innovative PrEP interventions" here. You need to have an end date March 5, 2020- WHEN?  Do this mean you are including innovations from the years 2023 and 2024. If so, those are probably less meaningful given you have chosen to center your paper on the pandemic. 

Generally, speaking, using PICO questions in this manuscript is not helpful. I would just state the inclusion criteria and then state the exclusion criteria (the latter of which you did). Using the PICO question in this paper takes up valuable word space.

There is no such thing as "Some of the exclusion criteria will include..." in a manuscript. Your exclusion criteria need to be specific and explicitly defined.

Why is the CDC (as opposed to the WHO) included in your search criteria when you are specifically focused on SSA PrEP guidelines etc?

Appendix A in conjunction with what is mentioned about this in the methods section is incomplete. Please provide all 10 questions that are being used to assess the quality of the review somewhere in the manuscript/Appendix A.

Where are you finding the operational managers who will be supplying the list of names of potential stakeholders? Are these individuals working in clinical organizations, governments, etc? How are the operational managers referring the potential stakeholders to you?

List the specific inclusion/exclusion criteria for stakeholders. What is currently mentioned is not sufficient.

Under step 3: " Each stakeholder will be asked to use mapping tools to score each other’s relationships on a point 10 scale as follows:" This does not make sense as it is currently written.

Overall, all of the steps in the stakeholder section need more explanation. I should be able to repeat every aspect of what you are doing based on what you have written. What is written currenly sounds like summaries.

Adjust formating of Table 2

It appears that during the qualitatitive interviews, you may only be recruiting from South Africa (based on the specific district mentioned in Table 2).  You need to go back to the background and even the beginning of the methods and provide clarity as to when and why you are recruiting from SSA vs South Africa.

Again, how will stakeholders be contacted (paper, emails, etc)? How iwll interviews be conducted (in-person vs virtual)?

The objective about implementation outcomes should be combined with the previous systematic review.

Discussion needs to be re-worked.

General: Please re-read and correct the numerous comma splices, split infinititives, and noun-verb tense disagreements.

Comments on the Quality of English Language

Introduction section needs some working. Many grammer errors here.

Author Response

Dear Reviewer,

Thank you for taking the time to review the manuscript, 

Attached please find the addressed comments as requested.

Best regards,

Reviewer 4 Report

Comments and Suggestions for Authors

The protocol presented is of great interest and relevance. The use of a mixed methods approach is appropriate and highly interesting to address the complexities of implementing PrEP interventions. 

Perhaps the fact of incorporating in the same protocol all the studies that the researchers want to conduct detracts from the concreteness and thoroughness of some questions:

- The introduction provides a good overview of the importance of PrEP among JAAJs and the context of its implementation during COVID-19. However, it would benefit from additional references to recent studies on the effectiveness and challenges of PrEP interventions globally and specifically in South Africa.

- Method 1:

The publication period of articles to be included in the review is unclear. We do not know how long the confinement in South Africa lasted.

- Method 2:

We are in the year 2024, how the authors are going to take into account that the recall of the actors may have been modified by the passage of time. It is even possible that some actors may no longer be in the resources from which they acted during the confinement of covid19.

- Method 3:

Include a more detailed explanation of the sample size and the rationale for choosing 20 actors for in-depth interviews.

That is, provide more specific criteria for the selection of participants, ensuring the inclusion of diverse perspectives within the actors involved in PrEP interventions.

Clearly define what is meant by ‘innovative interventions’. Each person may have a very different understanding of this.

- The authors point towards a process of triangulation but do not determine how this will be carried out. This would be a factor that would greatly enhance the quality of the work.

- Similarly, it would be relevant to add a section on the possible limitations of the study, including potential biases in participant selection and data collection, as well as methodological limitations.

Author Response

Dear Reviewer,

Thank you for taking the time to review the manuscript.

Attached please find the addressed comments as requested.

Best regards,

Round 2

Reviewer 3 Report

Comments and Suggestions for Authors

In the abstract, authors mention an "exploratory, ...multiple methods" design but in their response to the editor, they call there study an "explortory mixed methods design" and the also state mixed methods in line 91 and 92 of the methods section ("The proposed study will conduct an exploratory, descriptive design using a mixed -methods approach"). (You also mention mixed methods in your expected results section.) These terms are not the same and determining your exact type of study is very important.  If you do not plan on creating meta-inferences (that are explicitly stated in their own subsection) for your study strands, then you a have a "multiple or multi methods" study, not a "mixed methods" study. The authors need to look up the terminology and decide what type of study they have.  Reviewing the methodology differences between mixed methods and multiple/multi methods will inform them on if they need to add another section to this paper explicitly talking about how they will create metainferences. Additionally, there are specific concepts that are involved in mixed methods (see my initial questions from my first review of this manuscript referring to timing of strands, and which strands weigh most heavily).  Those questions actually need to be addressed specifically in the methods section of any paper that is mixed methods. If your paper is not mixed methods but is instead mult/multiple methods, then you don't need those terms and you need to use the appropriate language throughout. I will re-emphasize that mixed methods and multiple methods are not the same methodology. Refer to this citation (and others) to understand the difference- Creswell, J. (2015). Research Design: Qualitative, Quantitative and Mixed Methods Approaches. Pearson Education Inc.

Introduction- I appreciate the additional references; however the authors still only discuss 2 parts of the PrEP care continuum (uptake and adherence).  What about awareness, PrEP prescriptions, PrEP discussions, even showing up to PrEP referrals (if that's applicable in their healthcare system model)?   Use this article to share on some other areas of the PrEP Care Continuum https://www.ncbi.nlm.nih.gov/pmc/articles/PMC5333727/pdf/aids-31-731.pdf

I had made a previous comment regarding how the authors have linked their paper to the COVID-19 pandemic. I have read the authors response and feel that there may still be a misunderstanding.  As someone reading this paper, as it is currently phrased, it is heavy on the COVID-19 pandemic.   The authors have replied with the following: "Therefore, we aim to evaluate the innovative PrEP interventions implemented to prepare for future pandemics and improve the PrEP care continuum among AGYW since they have been prioritized by the Joint United Nations Programme on HIV/AIDS (UNAIDS) to achieve control of the HIV epidemic by 2030."  While this may be what they intend their paper to do, it does not read as such. Thus this is still a problem. Without adding in something about how the COVID pandemic impacted AGYWs perspectives, interests, etc in PrEP, this statement doesn't make sense and won't be as helpful in informing how future interventions.  To further my concern on this area, in response to my next comment, the authors have replied "The Specific comparison will be based on the PrEP interventions implemented before and during the pandemic which will enable the authors to assess the successes and challenges of these interventions." However, there still is now knowledge provided in the background section about how AGYW may or may not have changed in regards to their health, their behaviors, their health literacy, their hesitancy to medicine etc since the pandemic.

I appreciate the following additions "The emergence of the Coronavirus disease 2019 (Covid 19) pandemic with its treatment protocols, prevention measures and lock down requirements may have exacerbated the uptake and adherence of PrEP among AGYW [1 4]. However, guidelines for developing and implementing innovative interventions for service continuity were provided 61 by the World Health Organization (WHO) [ 1 5 ]."  What is missing are some brief examples of how the pandemic "exacerbated the update and adherence of PrEP among AGYW."  That is what is needed to make this make sense for readers who may not be in your community.

In response to the authors reply to my comment 3, I think there is still some confusion.  The authors need to make their case in the introduction as to why they focus one question on SSA and then everything else on South Africa. I am well aware that South Africa is part of SSA; however this statement " In 2022, AGYW accounted for more than 77% of new infections in Sub-Saharan Africa [2], whereas in South Africa they constituted about 37% of all new HIV infections [3]" by itself is not sufficient.  This is the only time I see both populations, the larger SSA and then more focused South Africa mentioned. From this statement, I do not know why the authors are interested in South Africa as opposed to SSA at large. As I mentioned in the first review "More comparison is needed in the background section to make this method make sense."

In response to their reply to comment 10, I appreciate the insight. I would actually briefly mention this insight in the paper as others will also wonder why the CDC was included in a global study.

In response to comment 11, thank you for that insight. It isn't able to be opened for me. I would just make sure the editors know that you want this link to be active in the publication.

Thank you for your response to comment 12.  Please add the following of your description of the people who are helping with recruitment- "public (government) primary healthcare facilities."  It is important for readers to know these individuals work for the government vs private vs some other type of employer as these things may speak to bias, access, and other key things someone may need to know when replicating aspects of your study. 

In regards to how are the operational mangers referring people to you, typically that means are these phone based, email, in-person, warm handoffs, etc?  That is the level of detail being requested.

Line 191--> gatekeepers is possessive plural (e.g. gatekeepers')

Response 16- thank you for your response, but this is not sufficient.  Similar to my earlier comments, in the background/introduction of your study, you need a statement that says PrEP uptake in SSA is low (or other concerns you have). In South Africa, PrEP uptake is also low, --> and then add some more details (not just the difference in HIV rates as the SSA rates are greater than South Africa).  From reading your methods section, it sounds like the reason you picked South Africa is because you have a relationship with local organizations. That's great, but that is not a scientific reason to do a study.  Your readers need to know why you are so much more concerned with South Africa compared to the rest of SSA. Are there unique structural or social factors in South Africa compared to the rest of SSA?  This is the information that needs to be explicitly stated in the intro to appropriately frame this study/set of studies.  You mention in the methods, that one of the districts has continued with HIV prevention services even during the pandemic.  This could be a reason why you focused on South Africa (if true). E.G. South Africa was one of the few countries that didn't shift all of their public health resources to pandemic needs and thus makes it an opportune country to study. Also, you mention this in the abstract "Although South Africa was the first country to register and roll out oral pre-exposure prophylaxis (PrEP) biomedical human immunodeficiency virus (HIV) prevention intervention in Sub-Saharan Africa (SSA), its uptake remains low, particularly among adolescent girls and young women (AGYW)."  It should also be mentioned in the background.  The abstract and the main manuscript are separate documents.  There is also a missed opportunity to specifically mention South Africa's barriers to PrEP uptake when you make the following statement: "Barriers to PrEP uptake include lack of access, low social support for users from peers and community, and inadequate information about PrEP [10 -12 ]"  Add the statement from the abstract and then elaborate on the specific South African barriers to PrEP uptake among AGYW specifically.  The barriers shouldn't be about South Africa in general or even SSA in general; they should be about South African AGYW. 

Re Comment 17, please see what I have written earlier about contact. For example, line 276 states "potential participants will be contacted..."  Will they be contacted by phone, email, in-person, etc?

There are several places you mention HIV-negative AGYW and PrEP (at least twice in your research questions).  This is a redudency as you can't be on PrEP and have HIV.  If this were the inclusion criteria, the wording would make sense; however in this context it can be remove

Author Response

Dear Reviewer,

Thank you for reviewing the manuscript,

All comments were addressed accordingly.

Best regards,
